# Genetic Variants in the *ABCB1* and *ABCG2* Gene Drug Transporters Involved in Gefitinib-Associated Adverse Reaction: A Systematic Review and Meta-Analysis

**DOI:** 10.3390/genes15050591

**Published:** 2024-05-07

**Authors:** Mariana Vieira Morau, Cecília Souto Seguin, Marília Berlofa Visacri, Eder de Carvalho Pincinato, Patricia Moriel

**Affiliations:** 1Department of Pharmacology, Faculdade de Ciências Médicas, Universidade Estadual de Campinas, Campinas 13083-888, SP, Brazil; marianavmorau@gmail.com (M.V.M.); ceciliaseguin@gmail.com (C.S.S.); 2Department of Pharmacy, Faculdade de Ciências Farmacêuticas, Universidade de São Paulo, São Paulo 05508-000, SP, Brazil; mariberlofa@gmail.com; 3Department of Clinical Pathology, Faculdade de Ciências Médicas, Universidade Estadual de Campinas, Campinas 13083-888, SP, Brazil; edercp@unicamp.br; 4Faculdade de Ciências Farmacêuticas, Universidade Estadual de Campinas, Campinas 13083-859, SP, Brazil

**Keywords:** gefitinib, *ABCB1*, *ABCG2*, adverse events, review, genetic variants

## Abstract

This systematic review and meta-analysis aimed to verify the association between the genetic variants of adenosine triphosphate (ATP)-binding cassette subfamily B member 1 (*ABCB1*) and ATP-binding cassette subfamily G member 2 (*ABCG2*) genes and the presence and severity of gefitinib-associated adverse reactions. We systematically searched PubMed, Virtual Health Library/Bireme, Scopus, Embase, and Web of Science databases for relevant studies published up to February 2024. In total, five studies were included in the review. Additionally, eight genetic variants related to *ABCB1* (rs1045642, rs1128503, rs2032582, and rs1025836) and *ABCG2* (rs2231142, rs2231137, rs2622604, and 15622C>T) genes were analyzed. Meta-analysis showed a significant association between the *ABCB1* gene rs1045642 TT genotype and presence of diarrhea (OR = 5.41, 95% CI: 1.38–21.14, I^2^ = 0%), the *ABCB1* gene rs1128503 TT genotype and CT + TT group and the presence of skin rash (OR = 4.37, 95% CI: 1.51–12.61, I^2^ = 0% and OR = 6.99, 95%CI: 1.61–30.30, I^2^= 0%, respectively), and the *ABCG2* gene rs2231142 CC genotype and presence of diarrhea (OR = 3.87, 95% CI: 1.53–9.84, I^2^ = 39%). No *ABCB1* or *ABCG2* genes were positively associated with the severity of adverse reactions associated with gefitinib. In conclusion, this study showed that *ABCB1* and *ABCG2* variants are likely to exhibit clinical implications in predicting the presence of adverse reactions to gefitinib.

## 1. Introduction

Gefitinib inhibits the tyrosine kinase (TKI) activity of the epidermal growth factor receptor (EGFR) [1]. It is considered to be a small molecule; therefore, it is administered orally. Gefitinib targets the adenosine triphosphate (ATP) molecules on the EGFR [2]. Gefitinib has yielded clinically positive responses in patients with advanced and metastatic non-small-cell lung cancer (NSCLC) [3] with hyperactive and mutated *EGFR* genes, primarily in exon 19 (del 19) or exon 21 (L858R) regions [4].

Previous studies have shown that gefitinib is well tolerated compared with other first- and second-generation TKIs, such as osimertinib and afatinib, respectively [5]. However, adverse drug reactions (ADRs) can potentially affect patient therapy and quality of life [6]. Commonly described gefitinib-associated ADRs include dermatological, gastrointestinal, and hepatic reactions [7].

Dermatological ADRs are the most reported (71–85%), with mild or moderate skin rashes being the most common. However, in approximately 18% of patients, these rashes can be severe, necessitating dose management or treatment suspension [8]. Additionally, hepatic toxicity can develop in these patients, with approximately 5–18% experiencing more severe forms (namely, grade 3, according to Common Terminology Criteria for Adverse Events (CTCAE) [9,10]). Diarrhea is a commonly associated gastrointestinal event that occurs in 18–95% of patients across all grades in the absence of antidiarrheal prophylaxis [11].

ADRs can seriously affect patient safety and quality of life and economic aspects of the associated healthcare system [12]. Part of the variability in reactions is explained by differences in drug metabolism, including the presence of enzymes of the Cytochrome P450 family (there are reports of hepatotoxicity and skin rash in individuals with *CYP2D6* enzyme polymorphisms) [13,14,15,16] and the ATP-binding cassette (ABC) family of drug transporters that catalyze the ATP-dependent active transport of chemically unrelated compounds, including anticancer drugs such as gefitinib [14,17]. These proteins are encoded by genes, and, therefore, there are possible genetic variations [18].

Gefitinib is a substrate for ABC efflux transporters that are highly expressed in the liver, intestine, brain, and tumor cells [17,19]. Reportedly, the ABC subfamily G member 2 (*ABCG2*) gene exhibits a high affinity for the TKI gefitinib [20]. Similarly, *ABCG2* genes (db/ID 1143C/T and 15622C/T) have been suggested to be involved in moderate and severe adverse diarrheal events [21]. Additionally, the ABC subfamily B member 1 (*ABCB1*) gene has been reported to be involved in gefitinib transport through the excretion of the metabolite O-desmethyl gefitinib via the bile pathway [22]. The *ABCB1* gene (db/ID 1128503 T/T) may be associated with a high degree of rash and adverse diarrheal events [23].

Therefore, investigating the association between single-nucleotide polymorphisms (SNPs) and gefitinib-treatment-associated adverse reactions is a valuable strategy for patients undergoing this line of therapy and may contribute to the advancement of personalized medicine. Herein, we explore the potential relationships between genetic variants of the drug transporter genes *ABCB1* and *ABCG2* and the adverse reactions associated with gefitinib treatment in patients with NSCLC.

## 2. Materials and Methods

This review followed the Preferred Reporting Items for Systematic Reviews and Meta-Analyses statement (PRISMA) 2020 checklist and reporting guideline [24]. The protocol is registered in the Open Science Framework (https://osf.io/6se4d, accessed on 21 August 2023).

### 2.1. Search Strategy

A comprehensive literature search was conducted to identify relevant studies published before 10 January 2024 in PubMed, Virtual Health Library/Bireme, Embase, Web of Science, and Cochrane Library databases. The full search strategy for all databases is presented in Appendix A. References found in the included studies were evaluated to include potential studies that had not yet been identified. The search was restricted to studies on humans and those published in English or Spanish languages. Duplicate studies were excluded from the analysis.

### 2.2. Study Selection

The PECOS (population, exposure, comparator, outcomes, and study design) model was used to select the potential studies: P (population): adult patients with NSCLC treated with gefitinib; E (exposure): the presence of genetic variants in drug transporters (namely, ABC transporters); C (comparator): absence of genetic variants of drug transporters (namely, ABC transporters); O (outcome): systemic adverse reactions (presence and severity of skin rash, diarrhea, and liver dysfunction); and S (study design): observational (namely, cohort, case–control, and cross-sectional) studies. Abstracts, preprints, studies using biobanks, and ecological studies were excluded.

Two reviewers (M.V.M. and C.S.S.) independently screened the titles and abstracts of citations to identify potentially relevant studies. Furthermore, they independently reviewed the obtained full-text articles according to the inclusion criteria. The third and fourth reviewers (M.B.V. and P.M.) resolved any disagreements. This process was performed using Rayyan [25], a free web application designed to help researchers conduct systematic reviews.

### 2.3. Data Extraction

The following data were extracted from the studies independently by two reviewers (M.V.M. and C.S.S.) using standardized sheets in Microsoft Excel: the name of the first author and publication year, country where the study was conducted, study design, sample size, male percentage, mean age, study population, gefitinib treatment (dose), toxicity assessment, toxicity evaluation, funding sources/sponsors, genotyping method, genes and SNPs studied, and the allelic frequency. Disagreements were resolved through discussion with the third and fourth reviewers (M.B.V. and P.M.).

### 2.4. Quality Assessment

The Newcastle–Ottawa Scale (NOS) [26] was used to evaluate the methodological quality of the studies (risk of bias) by two reviewers (M.V.M. and C.S.S.) independently, and disagreements were resolved by discussion with the third and fourth reviewers (M.B.V. and P.M.). Three primary domains were evaluated in each study, selection, comparability, and outcomes, and the maximum NOS scores for each domain were 4, 2, and 3 stars, respectively. Therefore, each study attained a total score of 9.

The Strengthening the Reporting of Genetic Association (STREGA) guidelines [27] were used for independent evaluation of the quality of genetic associations by two independent reviewers (M.V.M. and C.S.S.), and the disagreements were resolved by discussion with the third and fourth reviewers (E.C.P. and P.M.). The STREGA guidelines contain five main divisions as follows: genotyping methods and errors, population stratification, haplotype variation, Hardy–Weinberg equilibrium (HWE), and replication, with nine items to be evaluated in total. In all the studies, the total score was measured by assigning one point to each item. Better-quality studies scored higher.

### 2.5. Data Analysis

For the analysis of *ABCB1* gene polymorphisms, the following were assumed: for rs1128503 (1236C>T) and rs1045642 (3435C>T), the ancestral/wild allele was the C allele; for rs2032582 (2677G>T/A) and rs10256836, the ancestral/wild allele was the G allele. To analyze the genetic variants of the *ABCG2* gene, the following were assumed: for rs2231142 (421C>A), the ancestral allele is the C allele; for rs2231137 (34G>A), the ancestral allele is the G allele [28].

Statistical analyses were conducted using Review Manager (RevMan), version 5.4.1 (The Cochrane Collaboration, Oxford, UK). The data associations between the polymorphisms and presence of ADRs were conducted as follows: patients were divided into two groups based on their CTCAE grades [9]—those with grade 0 (without ADR) and those with grades ≥1 (with ADR). Furthermore, data associations between the polymorphisms and severity of ADRs were performed as follows: the patients were divided into two groups, those with grades 0 and 1 and those with a grade ≥2, according to the CTCAE [9]. Associations between genetic variants and ADR associated with gefitinib were calculated as pooled ORs and 95% CIs. Pooled OR was analyzed by the Mantel–Haenszel (M-H) method (fixed effect). The I^2^ test was used to determine heterogeneity. A range of 60–75% for I^2^ was considered significant, indicating substantial heterogeneity. An I^2^ of >75% represented considerable heterogeneity [29]. A *p*-value < 0.05 was considered statistically significant.

## 3. Results

### 3.1. Search Results

The electronic search identified 1503 potentially relevant studies. After removing duplicates and reviewing the titles and abstracts, 25 studies were selected for full-text analyses. Among these, five studies met the inclusion criteria [21,22,23,30,31]. No relevant studies were identified from the reference lists of the included studies. Figure 1 shows a flowchart of the literature search. The excluded studies and the exclusion criteria are detailed in Appendix A.

### 3.2. Characteristics of Studies, Adverse Drug Reactions, Genes/Genetic Variants, and Participants

The characteristics of the five studies are listed in Table 1. All studies were published between 2012 and 2021. Additionally, the studies included populations from different countries: participants from China (two studies), Japan (two studies), and the Netherlands (one study). All the studies exclusively included patients with NSCLC. The number of participants ranged from 31 to 184. Most of the study participants were female, and 56–68 years was the average age range. All the patients received 250 mg of gefitinib once daily.

All studies characterized the severity of ADRs using CTCAE; four studies used version 4.0, and one study used version 3.0. Five studies assessed dermatological ADRs, four evaluated gastrointestinal ADRs, three examined hepatic ADRs, and one examined interstitial lung disease. The studies included in this review differed greatly in the way the frequency of ADRs and their severity were reported according to the CTCAE.

In total, eight genetic variants of ABC transporter genes were reported, four of the *ABCG2* gene and four of the *ABCB1* gene. Regarding the *ABCG2* gene, rs2231142 (421C>A) appeared in four studies, and rs2231137 (34G>A) was studied in two studies, whereas rs2622604 (1143C/T) and 15622C/T were included in a single study. Regarding the *ABCB1* gene, rs10456242 (3435C>T), rs1128503 (1236C>T), and rs2032582 (2677 G>T/A) were observed in three studies and rs10256836 in only one study. Genotype frequencies in four studies followed the Hardy–Weinberg equilibrium (HWE), with one study not mentioned. Four of the five studies provided frequency data (absolute values or percentages) for genotypes observed in the studied populations. Polymerase chain reaction (PCR) was the most commonly used genotyping method. Peripheral blood samples were used for SNP analysis in all the studies. Table 2 provides detailed information on these aspects.

### 3.3. Adverse Drug Reactions versus ABCB1 and ABCG2 Genes

Diarrhea: Four studies investigated diarrhea, three of which obtained significant results. Ma et al. [23] studied the *ABCB1* rs1128503 genotype TT (*p* = 0.037, Fisher’s exact test; *p* = 0.011, dominant model) and rs10256836 genotype GG (*p* = 0.042, Fisher’s exact test; *p* = 0.024, co-dominant model), yielding significant results. Similarly, Kobayashi et al. [22] studied gene *ABCB1* rs2032582 and found significant differences in the TT + TA + AA genotype group (*p* = 0.032, Fisher’s exact test). The study by Lemos et al. [21] on the *ABCG2* 15622C/T gene was significant for the CC + CT genotype (*p* < 0.01) and *ABCG2* haplotype (*p* < 0.01).

Skin rash: All studies in this review explored dermatological ADRs; however, only Ma et al. [23] reported significant results relating to skin rash reactions, particularly for the *ABCB1* rs1128503 TT genotype (*p* = 0.015, Fisher’s exact test; *p* = 0.013, dominant model). Another study by Tamura et al. [31] reported a significant association between skin rashes and the *ABCG2* 34G>A genotype GG (*p* = 0.046).

Liver dysfunction: Liver-function-associated ADRs were studied in only three studies in this review, with one study showing significant results. Ma et al. [23] reported that the *ABCG2* gene rs2231142 was significant (*p* = 0.036, co-dominant model). Table 3 presents a detailed analysis of the results.

### 3.4. Quality Assessment

The methodological quality of the five studies based on the Newcastle–Ottawa Scale (NOS) is presented in Table 4. The total scores ranged from six to eight stars, with Guan et al. [30] obtaining the lowest score. Additionally, Kobayashi et al. [22], Lemos et al. [18], and Tamura et al. [31] obtained the highest scores. In the selection domain, only Guan et al. [30] did not receive a star for the exposure item, whereas other studies received five stars. In the comparability domain, which assesses confounding factors in case–control studies, no study received a star as they did not fit into this study type. In the outcome domain, Guan et al. [30] and Ma et al. [23] scored only two stars, whereas Kobayashi et al. [22], Lemos et al. [21], and Tamura et al. [31] scored three stars.

The quality of the included studies based on the Strengthening the Reporting of Genetic Association (STREGA) guidelines is shown in Table 5. The total scores ranged from five to eight points. The study by Guan et al. [30] was not included in this evaluation because it required genotype data for the application of the instrument. The highest score was achieved by Lemos et al. [21], whereas other studies scored five points. The study by Ma et al. [23] was the only study that did not score in the domain concerning the description of methods and errors in the genotyping process. Lemos et al. [21] and Ma et al. [23] mentioned haplotype variation modeling. The study by Tamura et al. [31] was the only one that did not consider the HWE.

### 3.5. Meta-Analysis Results

Finally, four of the five studies were selected for the meta-analyses [21,22,23,31]. The study by Guan [30] was not included because he did not provide genotype data. The meta-analysis was carried out in relation to two aspects, the presence and absence of adverse reactions (Figure 2, Figure 3 and Figure 4 and Appendix A) and the severity of adverse reactions (Appendix A). Genotype grouping, adverse reaction presence, and severity data are provided in Appendix A (grade ≥2 versus grade 0 + 1) and Appendix A (grade 0 versus ≥ 1).

Figure 2 illustrates the forest plot showing the association of the *ABCB1* gene rs/id1045642 with the polymorphism 3435C>T (CC group versus CT + TT group and TT group versus CC + CT) and the presence of cutaneous, gastrointestinal, and hepatic adverse reactions. This meta-analysis showed a genetic association (group TT versus group CT + CC) with the presence of ADR diarrhea (Figure 2d): odds ratio (OR) = 5.41; 95% confidence interval (CI): 1.38–21.14; *p* = 0.02; and heterogeneity (I^2^): 0%. The other meta-analyses for the skin rash (Figure 2a,b) events and liver dysfunctions (Figure 2e,f) showed no significant association with the CC group versus CT + TT and TT group versus CT + CC group in the *ABCB1* gene rs1045642. In Appendix A, it is possible to observe that there are no associations per allele (T allele versus C allele) at rs1045642 in the *ABCB1* gene with skin rash.

Figure 3 shows the forest plots associating the *ABCB1* gene (rs1128503) and ADR presence, namely, skin rash (Figure 3a,b), diarrhea (Figure 3c,d), and liver dysfunction (Figure 3e,f). For the skin rash ADR, the *ABCB1* gene rs1128503 with the polymorphism 1236C>T (CT + TT group versus CC group) was significantly associated with the presence of skin rash development (OR = 6.99; 95% CI: 1.61–30.30; *p* = 0,009; and I^2^ = 0%). Moreover, in the TT group versus CT + CC group, the appearance of skin rash was noted to be significant (OR = 4.37; 95% CI: 1.51–12.61; *p* = 0.006; and I^2^ = 0%). In Appendix A, it is possible to note the association of the *ABCB1* gene rs1128606 polymorphism 1236C>T with the T and C alleles and the presence of skin rashes (*p* < 0.00001). The subsequent meta-analyses for the diarrhea events and liver dysfunctions showed no significant association with the CC group versus CT + TT and TT group versus CT + CC group in the *ABCB1* gene rs1128503. 

Figure 4 shows the forest plots associating the *ABCG2* gene (rs2231142) and ADR presence (namely, skin rash, diarrhea, and liver dysfunction). For the skin rash ADR (Figure 4a), the CC group versus CA + AA group was not significantly associated with the presence of skin rash development (OR = 1.98; 95% CI: 0.88–4.48; *p* = 0.10; and I^2^ = 0%). The subsequent meta-analysis for diarrhea (Figure 4b) showed significant association with the CC versus CA + AA groups in the *ABCG2* gene rs2231142, with OR = 3.87; 95% CI: 1.53–9.84; *p* = 0,004; and I^2^ = 39. And, finally, for liver dysfunction (Figure 4c), the meta-analysis showed no significant association with the CC group versus CA + AA group. 

No *ABCB1* or *ABCG2* genes were positively associated with the severity of adverse reactions associated with gefitinib (Appendix A).

## 4. Discussions

To the best of our knowledge, this review is the first to compile studies linking ABC family transporter genes with gefitinib-associated adverse reactions in patients with NSCLC. This systematic review and meta-analysis aimed to identify potential associations between SNP-type polymorphisms in *ABCB1* and *ABCG2* genes and gefitinib-related ADRs. The systematic search yielded five studies, of which four were suitable for meta-analysis. Additionally, four studies showed significant associations between SNPs and gefitinib-associated ADRs. The methodological quality of the included genetic association studies was not considered high considering error rates, call rate descriptions, batch genotyping, and modeling of haplotype variation. However, the methodological quality of the studies was deemed high, indicating a low risk of bias.

Gefitinib was one of the first EGFR TKIs to be available on the pharmaceutical market for the treatment of EGFR-mutated NSCLC and is widely used [32]. Herein, it was observed that the primary gefitinib-associated adverse reactions were skin rashes, followed by diarrhea and hepatotoxicity, varying in severity from grade 1 to > grade 2 on the CTCAE scale. The assessment of adverse events was based on a standard instrument, although bias may be present because of differences in the assessment intervals. For instance, Tamura et al. [31] conducted evaluations at 2 months, whereas Kobayashi et al. [22] performed assessments after 14 days of TKI use.

Analyses of genetic variants revealed the relevance of the SNP rs1045642 (3435C>T) in the *ABCB1* gene, and it was examined in four out of the five studies included. ABC, P-glycoprotein (P-gp; multidrug resistance 1), and efflux transporters are important and of considerable clinical relevance, and studies have indicated their effects on gefitinib transport [33]. Our findings from the meta-analysis showed that the 3435C>T *ABCB1* polymorphism, specifically, the TT genotype, is associated with the presence of diarrhea, unlike the studies by Ma et al. [23] and Kobayashi et al. [22], which found no direct relationship between the genotype and the presence of this gastrointestinal reaction. This variant of the *ABCB1* gene is the synonymous C-to-T transition at nucleotide position 3435 located in exon 26 (rs1045642), which, despite not altering the amino acids, is markedly associated with reduced drug transport out of the membrane [34]. Although the 3435C>T polymorphism is silent, studies have shown that it can affect the evolution of proteins, consequently altering their function or substrate specificity [35]. 

Another relevant finding was related to the 1236C>T polymorphism, also in the *ABCB1* gene rs1128503, which was found to be positively associated with the presence of the dermatological skin rash reaction and the TT genotype. Findings regarding rs1128503 (1236C>T) reported in two studies are conflicting. Ma et al. [23] suggested that patients with the TT genotype may present a risk of developing rashes and diarrhea. However, Kobayashi et al. [22] found no association between the same genetic variants and gefitinib-associated ADRs. P-gp is expressed on the upper surface of the bronchial epithelium, and, in lung cancer, anticancer drugs such as platinum and etoposide can change its expression [36,37]. Thus, *ABCB1* polymorphisms may be associated with the accumulation of metabolites and drugs that may increase the risk of adverse events or disease development [38,39].

The *ABCG2* gene, also known as the breast cancer resistance protein, is related to a drug transport mechanism that is not fully understood [40]. However, studies have shown a notable impact on the pharmacokinetics of various compounds, such as lapatinib, a TKI used in treating metastatic and advanced breast melanomas [41]. Reportedly, in the intestine, *ABCG2* can reduce oral drug bioavailability and increase systemic exposure to anticancer drugs [42]. Herein, the most common *ABCG2* SNP observed was 421C>A (rs2231142), which is located in exon 5 and results in the alteration of the amino acid glutamine to lysine [43]. The *ABCG2* transporter not only acts as an efflux pump for drugs but also as a critical factor in inflammatory processes and autoimmune responses [44]. A review that delved into this polymorphism (421C>A) and its involvement in gefitinib showed evidence that it is a good predictor of skin toxicity but not a reliable marker [45]. Herein, we found that the *ABCG2* gene ID/rs2231142 (421C>A) SNP was mentioned in four studies; our meta-analysis study showed a close relationship between the CC genotype in rs2231142 and the presence of diarrhea, in line with the study by Lemos et al. [21], which reported a strong association with grade 2/3 diarrhea for the same *ABCG2* gene but with the 15622 C/T SNP, CC + CT genotype, and *ABCG2* haplotype. However, the report by Cusati et al. [46] showed a possible association between *ABCG*2 gene polymorphism 421C>A and the presence of diarrhea in patients treated with gefitinib.

This systematic review and meta-analysis have certain limitations. The number of included studies and the number of patients were relatively low. Some studies may have been missed because they were not indexed in the searched databases. Additionally, studies of interest that may have become available outside the established search period were excluded. Another notable limitation is genotype groupings when related to adverse reactions, which prevented the inclusion of other studies in our meta-analysis. It is well known that, in recent years, several TKIs have been reported for the treatment of NSCLC worldwide; however, in developing countries, TKIs that are often available to the population and in the health system are still first-generation inhibitors, such as gefitinib, which is why the emphasis is on the importance of further studies being carried out in association with poor populations [47,48].

## 5. Conclusions

Altogether, this systematic review indicates that genetic variants of *ABCB1* (rs1045642 and rs1128503) and *ABCG2* (rs2231142) transporters are likely to exhibit clinical implications relating to the presence of adverse reactions (skin rash and diarrhea) to gefitinib in patients with NSCLC. Our review included five studies, and four were suitable for meta-analysis. Therefore, more studies are needed to validate and replicate the findings of this study.

## Figures and Tables

**Figure 1 genes-15-00591-f001:**
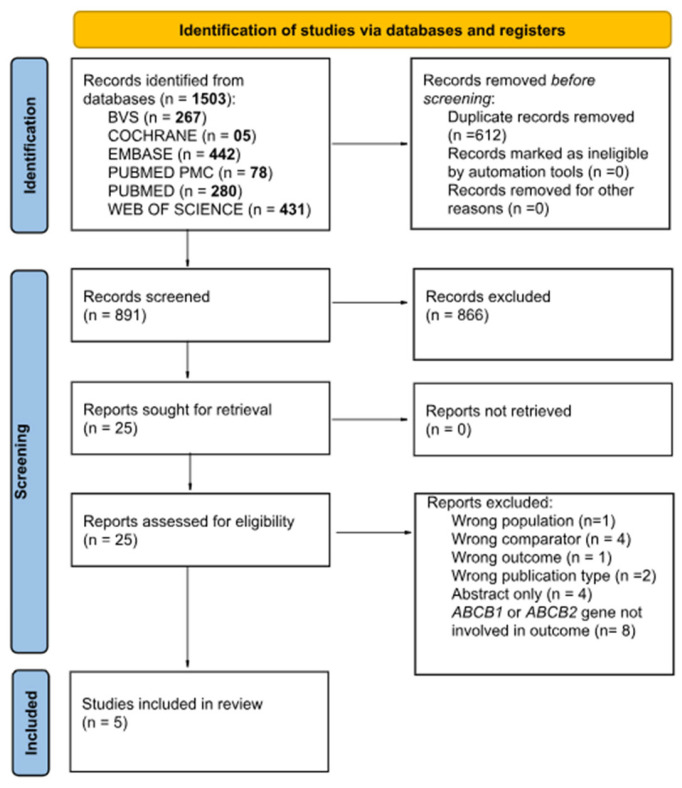
Study selection flowchart for literature search. *ABCB*, adenosine-triphosphate-binding cassette subfamily B member.

**Figure 2 genes-15-00591-f002:**
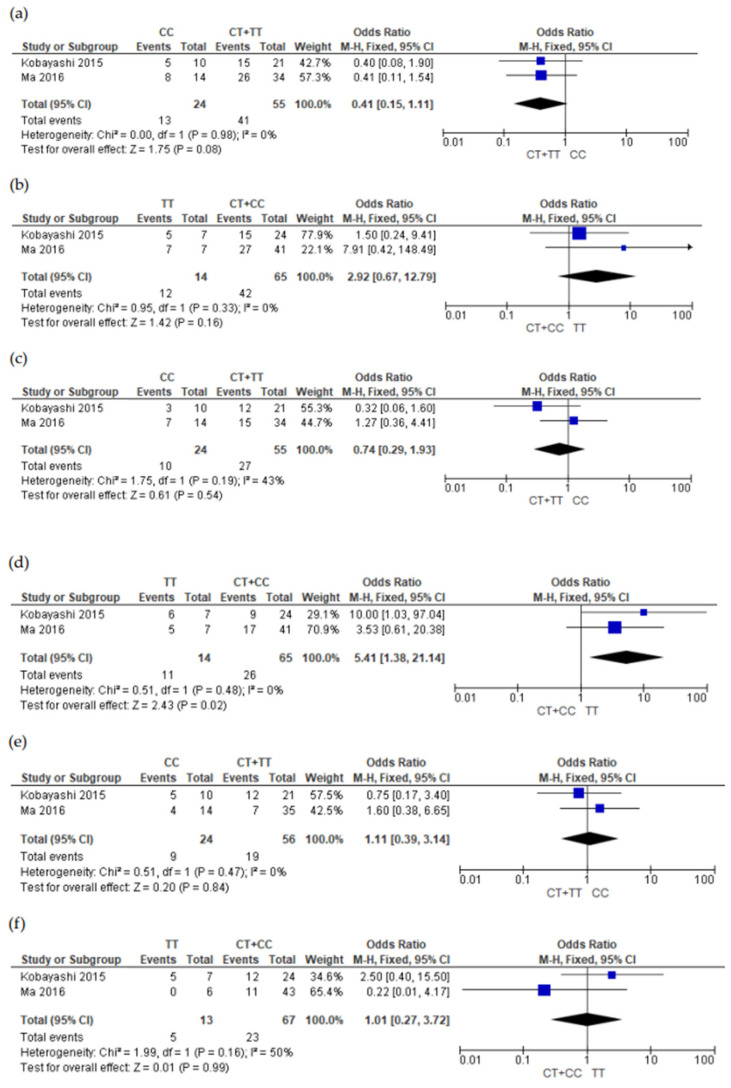
Forest plots of association between adenosine-triphosphate-binding cassette subfamily B member 1 (*ABCB1*) gene (rs1045642) genetic variant and adverse drug reaction (grade 0 versus grade ≥1). (**a**) Skin rash (CC versus CT + TT); (**b**) skin rash (TT versus CT + CC); (**c**) diarrhea (CC versus CT + TT); (**d**) diarrhea (TT versus CT + CC); (**e**) liver dysfunction (CC versus CT + TT); and (**f**) liver dysfunction (TT versus CT + CC) [22,23]. CI, confidence interval; M-H, Mentel-Haenszel; I^2^, heterogeneity; Chi^2^, Chi-square test for heterogeneity.

**Figure 3 genes-15-00591-f003:**
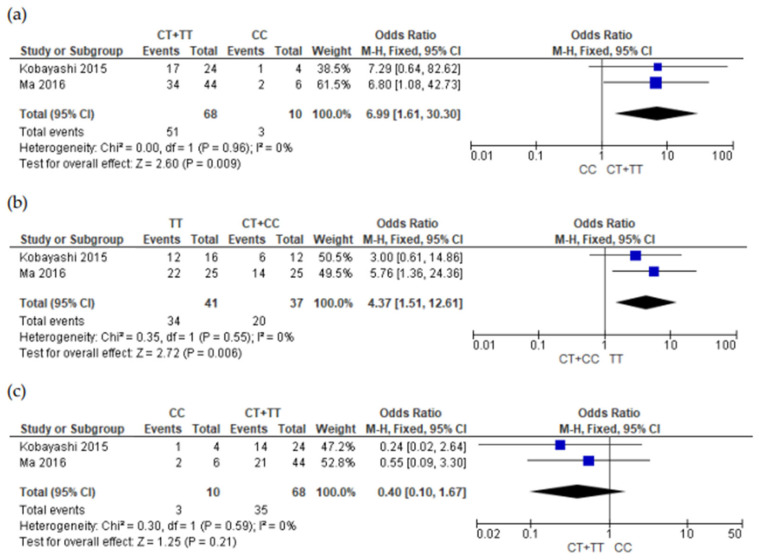
Forest plots of association between adenosine-triphosphate-binding cassette subfamily B member 1 (*ABCB1*) gene (rs1128503) genetic variant and adverse drug reaction (grade 0 versus grade ≥1). (**a**) Skin rash (CT + TT versus CC); (**b**) skin rash (TT versus CT + CC); (**c**) diarrhea (CC versus CT + TT); (**d**) diarrhea (TT versus CT + CC); (**e**) liver dysfunction (CC versus CT + TT); and (**f**) liver dysfunction (TT versus CT + CC) [22,23]. CI, confidence interval; M-H, Mentel-Haenszel; I^2^, heterogeneity; Chi^2^, Chi-square test for heterogeneity.

**Figure 4 genes-15-00591-f004:**
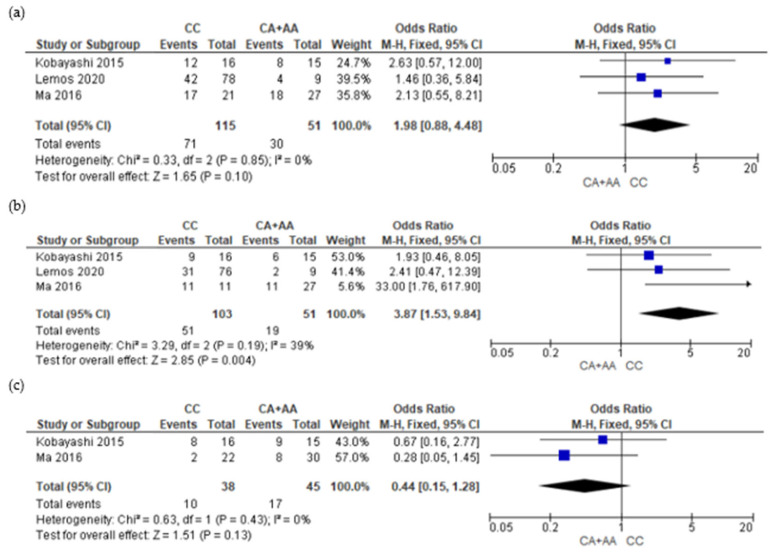
Forest association plots between adenosine-triphosphate-binding cassette subfamily G member 2 (*ABCG2*) gene (rs2231142) genetic variant and adverse drug reaction (grade 0 versus grade ≥1). (**a**) Skin rash (CC versus CA + AA); (**b**) diarrhea (CC versus CA + AA); and (**c**) liver dysfunction (CC versus CA + AA) [21,22,23]. CI, confidence interval; M-H, Mentel-Haenszel; I^2^, heterogeneity; Chi^2^, Chi-square test for heterogeneity.

**Table 1 genes-15-00591-t001:** Characteristics of included studies.

Author, Year	Country	Study Design	Sample Size (Male, %)	Age, Mean ± SD or Median (Range)	Population	Gefitinib Treatment	Adverse Reaction Assessment	Adverse Reaction Severity Frequency (Grade)	Funding Sources/Sponsors
Guan et al., 2021 [30]	China	Prospective (C)	184 (40.2)	NR	NSCLC patients	250 mg/ day	Skin rash	98 (0–2) 22 (3–4) ^B^	NA
Kobayashi et al., 2015 [22]	Japan	Prospective (C)	31 (41.9)	68 ± 8.6 (51–81)	NSCLC patients	250 mg/ day	Skin rash Diarrhea Liver dysfunction	9/11 (1/2) 10/4/1 (1/2/3) 12/1/3/1 (1/2/3/4) ^B^	NA
Lemos et al., 2020 [21]	Netherlands	Retrospective (C)	94 (56.4)	63.5	NSCLC patients	250 mg/ day	Skin rash Diarrhea	65/2 (0–1/2–3) 79/6 (0–1/2–3) ^A^	NA
Ma et al., 2017 [23]	China	Retrospective (CC)	59 (49.0)	56.0 (31–77)	NSCLC patients	250 mg/ day	Skin rash Diarrhea Liver dysfunction	18/13/1/4 (1/2/3/4) 19/4 (1/2) 6/2/2/1 (1/2/3/4) ^B^	NA
Tamura et al., 2012 [31]	Japan	Retrospective (C)	83 (42.0)	65.0 (36–86)	NSCLC patients	250 mg/ day	Skin rash Diarrhea Liver dysfunction ILD	23 (2–4) 4 (2–4) 15 (2–4) 5 (2–4) ^B^	NR

C: cohort study; CC: case–control study; NSCLC: non-small-cell lung cancer; ILD: interstitial lung disease; NR: not reported; NA: not applicable; SD: standard deviation. ^A^ denotes Common Terminology Criteria for Adverse Events version 3, and ^B^ denotes version 4.

**Table 2 genes-15-00591-t002:** Genetic variants studied in this systematic review.

Gene	Author, Year	Sample	Genotype Method	dbID/rs (Genetic Variants/ Polymorphism)	Frequency Genotype
*ABCB1*	Guan et al., 2021 [30]	Blood	Agena MassARRAY system	rs1128503 (1236C>T) rs2032582 (2677G>T/A)	NR
	Kobayashi et al., 2015 [22]	Blood	PCR-RFLP	rs1128503 (1236C>T) rs2032582 (2677G>T/A) rs1045642 (3435C>T)	C/C (*n* = 4) C/T (*n* = 8) T/T (*n* = 16) G/G (*n* = 3) G/T (*n* = 13) T/T (*n* = 6) T/A (*n* = 4) A/A (*n* = 1) C/C (*n* = 10) C/T (*n* = 14) T/T (*n* = 7)
	Ma et al., 2017 [23]	Blood	Sequenom MassARRAY system	rs1128503 (1236C>T) rs2032582 (2677G>T/A) rs1045642 (3435C>T) rs10256836 (C>G)	*Wt/Wt* (*n* = 25) *Wt*/m (*n* = 26) m/m (*n* = 8) Wt/Wt (*n* = 9) Wt/m (*n* = 23) m/m (*n* = 6) *Wt/Wt* (*n* = 19) *Wt/*m (*n* = 29) m/m (*n* = 7) *Wt/Wt* (*n* = 39) *Wt/*m (*n* = 18) m/m (*n* = 1)
	Tamura et al., 2012 [31]	Blood	Real-time PCR	rs1045642 (3435C>T)	CC (*n* = 23) CT (*n* = 44) TT (*n* = 16)
*ABCG2*	Kobayashi et al., 2015 [22]	Blood	PCR-RFLP	rs2231142 (421C>A)	C/C (*n* = 16) C/A + A/A (*n* = 15)
	Ma et al., 2017 [23]	Blood	Sequenom MassARRAY system	rs2231142 (421C>A) rs2231137 (34G>A)	Wt/Wt (*n* = 26) Wt/m (*n* = 25) m/m (*n* = 5) Wt/Wt (*n* = 25) Wt/m (*n* = 28) m/m (*n* = 4)
	Tamura et al., 2012 [31]	Blood	Real-time PCR	rs2231142 (421C>A) rs2231137 34G>A	CC (*n* = 45) CA (*n* = 31) AA (*n* = 7) GG (*n* = 51) GA (*n* = 28) AA (*n* = 4)
	Lemos et al., 2020 [21]	Blood or paraffin-embedded tumor sample	Real-time PCR	rs2231142 (421C>A) rs2622604 (1143C>T) 15622C/T	CC (*n* = 83) CA (*n* = 10) AA (*n* = 1) CC (*n* = 54) CT (*n* = 34) TT (*n* = 3) CC (*n* = 47) CT (*n* = 35) TT (*n* = 7)

PCR: polymerase chain reaction; Wt: wild type; Wt/m: heterozygous type; m: mutant type; PCR-RFLP: polymerase chain reaction restriction fragment length polymorphism; NR: not reported; *ABCB1*: adenosine-triphosphate-binding cassette subfamily B member 1; and *ABCG2*: adenosine-triphosphate-binding cassette subfamily G member 2.

**Table 3 genes-15-00591-t003:** Association between adenosine-triphosphate-binding cassette (ABC) subfamily B member 1 (*ABCB1*) and ABC subfamily G member 2 (*ABCG*2) genetic variants and adverse reactions studied in this systematic review.

Gene	dbID/rs (Genetic Variants/ Polymorphism)	ADRs (Number of Patients)						
		Skin Rash		Diarrhea		Liver Dysfunction		Author, Year
		Grades 1–4 or 2+	Association Significative?	Grades 1–4 or 2+	Association Significative?	Grades 1–4 or 2+	Association Significative?	
* **ABCB1** *	rs1045642 (3435C>T)	7 (CC) 16 (CT + TT) ^b^	No	1 (CC) 3 (CT + TT) ^b^	No	3 (CC) 12 (CT + TT) ^b^	No	Tamura et al., 2012 [31]
		8 (CC) 19 (CT) 7 (TT) ^a^	Yes **	6 (CC) 8 (CT) 4 (TT) ^a^	No	7 (CC) 10 (CT) 5 (TT) ^a^	No	Ma et al., 2017 [23]
		5 (CC) 10 (CT) 5 (TT) ^a^	No	3 (CC) 6 (CT) 6 (TT) ^a^	No	5 (CC) 7 (CT) 5 (TT) ^a^	No	Kobayashi et al., 2015 [22]
	rs1128503 (1236C>T)	2 (CC) 12 (CT) 22 (TT) ^a^	Yes *	2 (CC) 5 (CT) 16 (TT) ^a^	Yes *	3 (CC) 4 (CT) 4 (TT) ^a^	No	Ma et al., 2017 [23]
		1 (CC) 5 (CT) 12 (TT) ^a^	No	1 (CC) 6 (CT) 8 (TT) ^a^	No	1 (CC) 5 (CT) 9 (TT) ^a^	No	Kobayashi et al., 2015 [22]
	rs2032582 (2677G>T/A)	4 (GG) 14 (GT) 8 (TT) ^a^	No	4 (GG) 11 (GT) 1 (TT) ^a^	No	0 (GG) 7 (GT) 2 (TT) ^a^	Yes **	Ma et al., 2017 [23]
		1 (GG) 6 (GT) 8 (TT + TA + AA) ^a^	No	0 (GG) 9 (GT) 9 (TT + TA + AA) ^a^	No	1 (GG) 8 (GT) 6 (TT + TA + AA) ^a^	No	Kobayashi et al., 2015 [22]
	rs10256836 (C>G)	0 (CC) 7 (CG) 28 (GG) ^a^	No	0 (CC) 2 (CG) 20 (GG) ^a^	Yes *	0 (CC) 4 (CG) 7 (GG) ^a^	No	Ma et al., 2017 [23]
	rs1045642 rs1128503 rs2032582	16 (TTT) 10 non (TTT) ^a^	No	7 (TTT) 9 non (TTT) ^a^	^a^ No	5 (TTT) 4 non (TTT) ^a^	No	
* **ABCG2** *	rs2231142 (421C>A)	14 (CC) 9 (CA + AA) ^b^	No	3 (CC) 1 (CA + AA) ^b^	No	8 (CC) 7 (CA + AA) ^b^	No	Tamura et al., 2012 [31]
		17 (CC) 14 (CA) 4 (AA) ^a^	No	11 (CC) 8 (CA) 3 (AA) ^a^	No	2 (CC) 8 (CA) 0 (AA) ^a^	Yes *	Ma et al., 2017 [23]
		12 (CC) 8 (CA + AA) ^a^	No	9 (CC) 6 (CA + AA) ^a^	No	8(CC) 9(CA + AA) ^a^	No	Kobayashi et al., 2015 [22]
		63 (CC) 5 (CA + AA) ^a^	No	37 (CC) 2 (CA + AA) ^a^	No	N.A.	N.A.	Lemos et al., 2020 [21]
	rs2231137 (34G>A)	10 (GG) 13 (GA + AA)^b^	Yes *	2 (GG) 2 (GA + AA) ^b^	No	10 (GG) 15 (GA + AA) ^b^	No	Tamura et al., 2012 [31]
		13 (GG) 18 (GA) 4 (AA) ^a^	No	9 (GG) 11 (GA) 2 (AA) ^a^	No	3 (GG) 8 (GA) 0 (AA) ^a^	Yes **	Ma et al., 2017 [23]
	rs2622604 (1143C>T)	68 (CC + CT) 0 (TT) ^a^	No	39 (CC + CT) 0 (TT) ^a^	No	N.A.	N.A.	Lemos et al., 2020 [21]
	15622C/T	59 (CC + CT) 6 (TT) ^a^	No	30 (CC + CT) 6 (TT) ^a^	Yes *	N.A.	N.A.	
	haplotype 1143C/T and 15622 C/T	6 (TT-TT + TT-other) 59 (other–other) ^a^	No	6 (TT-TT + TT–other) 30 (other–other) ^a^	Yes *	N.A.	N.A.	

ADR: adverse drug reaction; *ABCB1*: adenosine-triphosphate-binding cassette subfamily B member 1; and *ABCG2*: adenosine-triphosphate-binding cassette subfamily G member 2. ^a^: grades 1–4; and ^b^: grade +2. *: yes, association significant: *p* > 0.05; **: yes, association significant: *p* > 0.10; and N.A.: not applicable.

**Table 4 genes-15-00591-t004:** Methodological quality of the studies included based on the Newcastle–Ottawa Scale.

Author, Year	Selection				Comparability	Outcome/Exposure			Total Score
	Item 1	Item 2	Item 3	Item 4	Item 1	Item 1	Item 2	Item 3	
Guan et al., 2021 [30]	**	*		*		*	*		6
Kobayashi et al., 2015 [22]	**	*	*	*		**	*		8
Lemos et al., 2020 [21]	**	*	*	*		**	*		8
Ma et al., 2017 [23]	**	*	*	*		*	*		7
Tamura et al., 2012 [31]	**	*	*	*		**	*		8

Notes: Selection—Item 1: representativeness of the exposed cohort; Item 2: selection of the non-exposed cohort; Item 3: ascertainment exposure; and Item 4: demonstration that the outcome was not present at the start of the study. Comparability—Item 1: comparability of cohorts. Outcome/Exposure—Item 1: assessment of outcome; Item 2: follow-up of outcomes; and Item 3: adequacy of follow-up of cohorts. The total scores ranged from six to eight stars. * one star; ** two stars.

**Table 5 genes-15-00591-t005:** The quality of reporting using the Strengthening the Reporting of Genetic Association guidelines.

Author, Year	Description of Genotyping Methods and Errors					Description of Modeling Population Stratification?	Description of Modeling Haplotype Variation?	Hardy–Weinberg Equilibrium Was Considered?	Statement of Whether the Study Is the First Report of a Genetic Association, a Replication Effort, or Both?	Score
	Genotyping Methods and Platforms?	Error Rates and Call Rates?	Genotyping in Batches?	Laboratory/Center Where the Genotyping Was Performed?	The Numbers of Individuals Were Successful Genotyped?					
Kobayashi et al., 2015 [22]	No	No	Yes	No	Yes	Yes	No	Yes	Yes	5
Lemos et al., 2020 [21]	Yes	No	Yes	No	Yes	Yes	Yes	Yes	Yes	7
Ma et al., 2017 [23]	No	No	No	No	Yes	Yes	Yes	Yes	Yes	5
Tamura et al., 2012 [31]	Yes	No	Yes	No	Yes	Yes	No	No	Yes	5

Note: The total scores ranged from five to eight stars.

## Data Availability

The original contributions presented in the study are included in the article/Appendix A, further inquiries can be directed to the corresponding author.

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
