# Peer review of "Genetic Variants in the ABCB1 and ABCG2 Gene Drug Transporters Involved in Gefitinib-Associated Adverse Reaction: A Systematic Review and Meta-Analysis"

_genes, 2024, doi:10.3390/genes15050591_

Round 1
Reviewer 1 Report
Comments and Suggestions for Authors
In general, the meta-analysis presented is well designed and carried out. The main limitation, also noted by the authors, is the low number of studies included in the meta-analysis. However, the results are interesting, although more studies are needed in the future to corroborate them.
There are some questions that I would like the authors to answer, and I will detail them below.
1. Why is the rs31137 polymorphism of the ABCG2 gene evaluated by the two studies included in the meta-analysis (Ref #20 and #28) not included in the analysis? This should be clarified.
2. All genes must be italicized.
3. In Table 2, the "M" of the genotypes must be lowercase, according to what is stated in the abbreviations to the footnote of the table.
4. The footnote of the table 4 must include the phrase: “The total scores ranged from six to eight stats”.
5. The footnote of the table 5 must include the phrase: “The total scores ranged from five to eight stats”.
Reviewer 2 Report
Comments and Suggestions for Authors
Authors of the manuscript presents an interesting review and meta-analysis on the association between the genetic variants of ABCB1 and ABCG2 genes and adverse reactions of gefitinib. Methodology and data analysis seem to be performed properly. Overall, the paper is well written and well-organized.
Minor remarks:
1. Introduction, lines 52-58. As the Authors stated, the variability in ADRs might be explained by several factors including differences in gefitinib metabolizing enzymes and drug transporters. Please mention which enzymes engaged in gefitinib metabolism are polymorphic and how it affects ADRs occurrence.
2. Table 2 should be corrected:
- specify methods for genotype detection in case of study by Ma et al., Tamura et al., and Lemos et al.;
- add genetic variants/polymorphism for rs10256836 (Ma et al.);
- in Table 2 legend “m” should be replaced by “M” as mutant type.
